# Sequentially activated discrete modules appear as traveling waves in neuronal measurements with limited spatiotemporal sampling

Yuval Orsher[1,2], Ariel Rom[1,3], Rotem Perel[1], Yoav Lahini[2,3], Pablo Blinder[1,3], Mark Shein-Idelson[1,3]*

[1]School of Neurobiology, Biochemistry, and Biophysics, Tel Aviv University, Tel Aviv, Israel; [2]School of Physics & Astronomy, Faculty of Exact Sciences, Tel Aviv University, Tel Aviv, Israel; [3]Sagol School of Neuroscience, Tel Aviv University, Israel, Tel Aviv, Israel

**\*For correspondence:**
sheinmark@tauex.tau.ac.il

**Competing interest:** The authors declare that no competing interests exist.

**Abstract** Numerous studies have identified traveling waves in the cortex and suggested they play important roles in brain processing. These waves are most often measured using macroscopic methods that are unable to assess the local spiking activity underlying wave dynamics. Here, we investigated the possibility that waves may not be traveling at the single neuron scale. We first show that sequentially activating two discrete brain areas can appear as traveling waves in EEG simulations. We next reproduce these results using an analytical model of two sequentially activated regions. Using this model, we were able to generate wave-like activity with variable directions, velocities, and spatial patterns, and to map the discriminability limits between traveling waves and modular sequential activations. Finally, we investigated the link between field potentials and single neuron excitability using large-scale measurements from turtle cortex ex vivo. We found that while field potentials exhibit wave-like dynamics, the underlying spiking activity was better described by consecutively activated spatially adjacent groups of neurons. Taken together, this study suggests caution when interpreting phase delay measurements as continuously propagating wavefronts in two different spatial scales. A careful distinction between modular and wave excitability profiles across scales will be critical for understanding the nature of cortical computations.

## eLife assessment

In this **important** work, authors show that brain activity thought to be a travelling wave may just be a series of sequentially activated sources at the neuron spiking level. They support this with **convincing** results from a turtle cortex preparation and relevant simulations. This work will be of interest to neuroscientists interested in understanding how cortical computations are made.

## Introduction

Cortical traveling waves are considered a ubiquitous feature of cortical dynamics and are suggested to play important roles in cortical processing (*Muller et al., 2018*; *Muller and Destexhe, 2012*; *Sato et al., 2012*; *Wu, 2008*). These waves are observed across a wide range of spatial and temporal scales and exhibit a wide repertoire of propagation patterns. Traveling waves can propagate to specific directions (plane wave) (*Lubenov and Siapas, 2009*; *Massimini et al., 2004*; *Senseman, 1999*; *Xu et al., 2007*), outward from a center locus (radial waves) (*Muller et al., 2014*) or rotate around a phase singularity (spiral

waves) (*Huang et al., 2010*; *Muller et al., 2016*; *Prechtl et al., 1997*). Waves also range in their spatial extent from mesoscale waves spanning a single cortical area (*Ferezou et al., 2006*; *Jancke et al., 2004*; *Lubenov and Siapas, 2009*; *Petersen et al., 2003*; *Reimer et al., 2011*; *Rubino et al., 2006*; *Takahashi et al., 2015*; *Xu et al., 2007*), to macroscale waves propagating through multiple brain areas and even across the two hemispheres (*Alexander et al., 2013*; *Goldman et al., 1949*; *Massimini et al., 2004*; *Muller et al., 2016*; *Patten et al., 2012*). Finally, cortical waves are characterized by different propagation velocities (from millimeters to meters per second) and frequency bands. For example, α-band oscillations (8–12 Hz) travel from occipital to frontal areas (*Hughes, 1995*); γ-band (30–80 Hz) auditory-induced oscillations travel from anterior to posterior (*Ribary et al., 1991*); and slow wave oscillations (0.1–1 Hz), occurring during sleep, travel from anterior to posterior cortical regions (*Massimini et al., 2004*).

It is unclear how the abundance of traveling waves across large areas of cortex fits its modular structural and functional organization (*Meunier et al., 2010*; *Sporns and Betzel, 2016*). Tangential cortical connectivity is not uniform. It is higher within cortical areas (e.g. V1) or regions (e.g. visual cortex) than between areas (V1 and V2 or visual cortex and auditory cortex) (*Harris et al., 2019*; *Hilgetag et al., 2000*; *Jia et al., 2022*; *Markov et al., 2011*). This modularity is observed across multiple scales. Within single cortical areas, neurons are segmented into columns with high intra-column connectivity and sparser inter-column connectivity (*Mountcastle, 1997*). Such non-uniform connectivity is likely to restrict the spread of activity rather than to support smooth propagation of waves across the cortical sheet during which action potentials are sequentially activated in neurons as the wavefront passes. Indeed, it was suggested that one of the roles of modularity is to segregate circuits and computation (*Jia et al., 2022*; *Shein-Idelson et al., 2016*) and prevent uncontrolled spread of excitability, as is evident in pathological conditions such as epileptic seizures (*Martinet et al., 2017*).

Understanding in which scenarios traveling waves ignore cortical modularity requires defining the spatial granularity with which waves are examined and performing experimental measurements with the corresponding resolution. For example, determining whether a traveling wave entails a spreading excitation at the single neuron level requires single cell resolution measurements. In contrast, traveling waves are commonly measured with low spatial resolution technique (*Hindriks et al., 2014*; *Mak-McCully et al., 2015*; *Muller et al., 2016*; *Ribary et al., 1991*). This is especially true for extracranial recording such as EEG in which brain current sources are distant from the measurement electrodes and separated by diffracting elements such as the skull (*Michel and Murray, 2012*). Even in more local intracranial measurements such as electrocorticography (ECoG), volume conduction through low impedance media, such as the cerebrospinal fluid, may result in substantial contribution of distant areas to local measurements and effective spatial low-pass filtering that blur out possible spatially segregated activations (*Buzsáki et al., 2012*; *Herreras, 2016*; *Kajikawa and Schroeder, 2011*). Imaging waves with more local reporters of excitability such as voltage sensitive dye (VSD) reduces the limitations imposed by large electrodes. While the spatial resolution of this method is higher, VSD measurements mostly reflect sub-threshold potentials (*Chemla and Chavane, 2010a*) with strong contribution from upper layers and long-range inputs (*Chemla and Chavane, 2010b*; *Petersen et al., 2003*; *Stepanyants et al., 2009*). Thus, VSDs integrate inputs from both local and distant spiking activities rather than represent local spiking excitability profiles. In addition, VSD signals may be contaminated by contribution from non-neuronal sources (*Pál et al., 2015*). For example, the velocities of some traveling waves correspond to axonal propagation (*Muller and Destexhe, 2012*). However, such axonal propagations do not necessarily activate the neurons in the target tissue and single cell excitability may remain modular. While in specific cases the existence of traveling waves in single cells was measured in high resolution (*Lubenov and Siapas, 2009*; *Voufo et al., 2023*), this existence cannot be unambiguously generalized to all reported traveling waves, especially given the large diversity of reported wave patterns across different brain areas.

In addition to the lack of understanding of the nature of excitability underlying waves, previous studies raised the possibility that measured propagating patterns may be biased by the limitations of experimental technique used (*Muller et al., 2014*; *Wu, 2008*; *Zhigalov and Jensen, 2023*). For example, when considering a scenario in which the intensity of a stationary source with a spatially decaying profile increases over time. In this scenario, additional regions cross the detection threshold at progressively longer times and may result in a false measurement of a propagating wave. To address this caveat, Muller et al. introduced the phase latency map (PLM) allowing them to discriminate between a stationary bump and a traveling wave and to conclude that propagating waves were

measured (*Muller et al., 2014*). While PLM address this specific caveat, measured traveling wave-like patterns (WLPs), that are not underlined by spatially continuous propagation, can be found in additional scenarios. For example, it was recently shown that a mixture of stationary sources can result in WLPs in EEG recordings (*Zhigalov and Jensen, 2023*). Similarly, current source density reveals WLPs following sequential activation of different cortical lamina (*Orczyk et al., 2021*). These studies raise important concerns about the propagation profiles in some systems. Despite these concerns, the criteria for differentiating WLPs from traveling waves remains unknown. Further, evidence for the existence of WLPs on the single neuron level is missing. Thus, we lack a classification of different identified traveling waves in terms of their underlying microscale excitability patterns.

Here, we use large-scale spiking measurements and simulations to examine the possibility that sequential activation of adjacent modules can be falsely identified as a traveling wave. Specifically, we first illustrate, using EEG simulations, that sequentially activating two adjacent cortical areas results in phase latency profiles resembling traveling waves. To link single neurons activation profiles with field potentials, we utilize the ex vivo eye-brain preparation of turtle cortex in which traveling waves have been previously reported in response to visual stimulation (*Prechtl et al., 1997*; *Senseman and Robbins, 2002*; *Shein-Idelson et al., 2017*). Using dense multi-electrode array (MEA) recordings, we compare local spiking activity with local field potentials (LFP) over several mm² of cortex. We show that while low-frequency oscillations in LFP appear to propagate as a smooth traveling wave, the underlying spiking activity is better described by a sequential activation of separate cortical populations. Finally, using an analytical model, we map the conditions for identifying sequential stationary activations as traveling waves and describe the high diversity of resulting WLPs.

## Results

### Sequential activation of two discrete brain areas appears as traveling waves in simulated EEG PLM

Traveling waves are commonly reported in EEG recordings (*Muller et al., 2018*) but the cortical activation profiles underlying these waves remain unclear. We used simulations to test if sequential but spatially stationary activations can appear as traveling waves in EEG recordings. Specifically, we simulated the sequential activation (with time delay of $\Delta T$) of two separate cortical areas in the parietal lobe (*Figure 1A*) and calculated the scalp EEG signals emanating from these activations using a forward model implemented in 'Brainstorm' (*Tadel et al., 2011*) (Methods). We modeled signal sources in each area as a set of randomly distributed and periodically activated dipoles (*Figure 1A*).

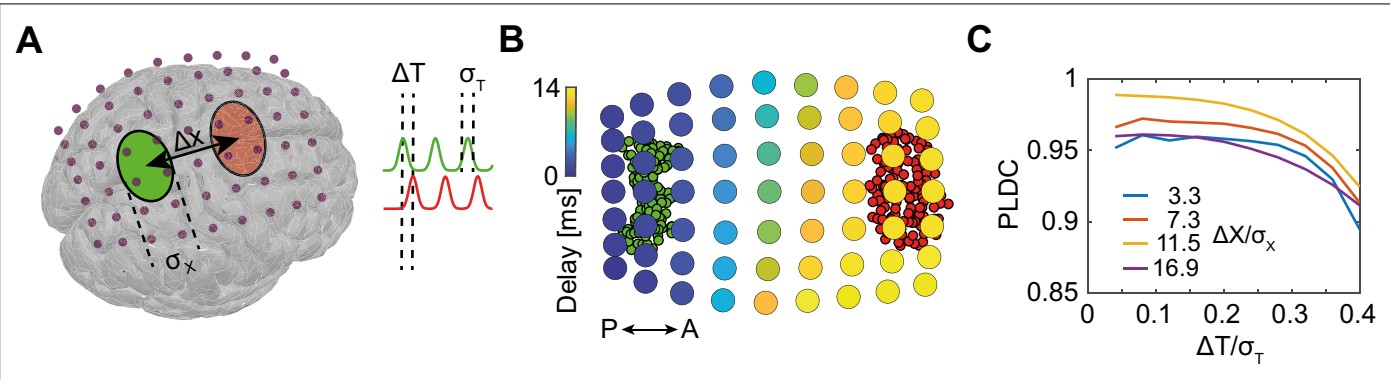

**Figure 1.** Sequential activation of two discrete brain areas appears as waves in simulated EEG phase latency maps. (**A**) Schematic illustration of simulated EEG. Two brain areas (green and red ellipses) are defined on the cortex by a set of dipoles that are activated uniformly with a time delay ($\Delta T$) between areas. A finite element forward model in Brainstorm (*Tadel et al., 2011*) is used to simulate the resulting potentials measured with EEG electrodes (purple dots). (**B**) Latency to peak measured across electrodes following the sequential activation of two sets of dipoles (green and red circles in the background) with a distance of 11 cm ($\Delta x/\sigma_x$=16.9) and time delay of 20 ms ($\Delta T/\sigma_T$=0.2). A continuous delay pattern propagating from posterior (P) to anterior (A) is observed. (**C**) Phase latency distance correlations (PLDCs) for different distances and different delays between two activated areas. PLDC remains close to 1 (indicating apparent wave propagation) for a wide range of values.

The online version of this article includes the following figure supplement(s) for figure 1:

**Figure supplement 1.** Sequential activation of two discrete small brain areas appears as waves in simulated EEG phase latency maps.

We first calculated the phase latency of the EEG signals across electrodes (*Figure 1B*). Interestingly, we observed a continuous range of delays gradually increasing along the anterior-posterior axis. Thus, phase delays did not reflect the modular construction of two spatially restricted sources. Rather, continuous propagation was observed for both large (*Figure 1B*) and small sources (*Figure 1—figure supplement 1A and B*). A standard approach for quantifying the existence of traveling waves is computing phase latency distance correlation (PLDC) (*Huang et al., 2010*; *Muller et al., 2016*; *Muller and Destexhe, 2012*; *Zhang et al., 2018*) which is the linear correlation between a defined phase crossing time (e.g. peak time) and the spatial location along the wave's trajectory. Applying this calculation to the data in *Figure 1B* along the posterior-anterior axis resulted in a value of 0.98, indicative of wave propagation. We next examined if such WLPs are also detected if the distance between areas or the time delay between activations changes. Repeating the simulations for a range of distances and delays (*Figure 1C*) revealed high PLDC values in all distances which only slowly decay with time delay. Furthermore, filtering the measured EEG signals with a narrow-band filter (a common practice when analyzing specific oscillatory bands; *Muller et al., 2018*; *Wu, 2008*) further increased PLDC (*Figure 1—figure supplement 1C and D*). These results suggest that detection of WLPs following sequential activation of spatially localized modules is abundant across parameters.

## WLPs in LFP are underlined by sequential activation of spatially discrete neuronal populations in MEA recordings from turtle cortex

Our next aim was to examine whether WLPs exist in cortical tissue. To do this, one needs a system with identified traveling waves across cortex, in which one can measure the microscopic tissue suprathreshold excitability, i.e., spikes. To achieve such measurements, we utilized the ex vivo eye-brain turtle preparation (*Shein-Idelson et al., 2017*). In this preparation (*Figure 2A*), the whole brain with the eye attached is extracted, and the dorsal cortex – a three-layered homolog of the mammalian cortex (*Tosches et al., 2018*) – is flattened onto a MEA. The advantage of this preparation is that it allows large-scale two-dimensional (in directions tangential to the cortical surface) sampling of both spiking and LFP activity. Furthermore, previous works with this preparation have shown the existence of waves that travel across cortex following visual stimulation (*Prechtl et al., 1997*; *Senseman and Robbins, 2002*). Finally, while the layered organization of turtle cortex is different than that of mammalian cortex, the basic excitability features of both tissues are similar (*Connors and Kriegstein, 1986*; *Hemberger et al., 2019*; *Kriegstein and Connors, 1986*; *Larkum et al., 2008*; *Shein-Idelson et al., 2017*), and substantial differences in the manner by which field potentials and spikes spread through the tissue are not to be expected.

Corresponding with previous reports (*Prechtl et al., 2000*), visual stimulation elicited increases in LFPs and spikes (*Figure 2B* – gray and black traces respectively) across large regions of the visual cortex. Although responses varied in intensity between stimulations, they typically started ~200 ms post-stimulus and lasted for ~2 s. Spectral analysis revealed that responses were characterized by high power at low frequencies (*Figure 2C*). Specifically, applying a 2 Hz low-pass filter captured the strong fluctuations in LFP following visual stimulations (*Figure 2B* – blue trace). We next set to examine how this low-frequency band propagates across cortex. Averaging over all trials revealed a clear propagation pattern traveling from anterolateral to posteromedial (*Figure 2D*). This pattern was consistent with previous reports of waves traveling from area D2 to area D1 in turtle cortex (*Prechtl et al., 1997*). To illustrate this propagation on the time domain, we plotted the first oscillation cycle of the filtered LFP (2 Hz low-pass, *Figure 2B* – red and blue dotted line) along a linear wave path (*Figure 2E*). The maximum showed a clear propagation pattern with a velocity of 1.8±0.35 mm/s (mean ± sd).

To quantitatively estimate the propagation trajectories during single oscillations cycles, we used the analytic signal of the Hilbert transform (*Le Van Quyen et al., 2001*; *Muller et al., 2014*; *Rubino et al., 2006*) (Methods). We first identified the $\pi/2$ phase crossing times of all oscillation cycles. We then grouped phase crossings into events belonging to single oscillation cycles based on continuity of phases across space and time (Methods, *Figure 2—figure supplement 1*). The first oscillation cycle showed a clear propagating wavefront, but more importantly, intense spiking activity. We therefore focused our analysis on this cycle and calculated its propagation direction. To do so, we defined the wave center path, as the weighted spatial center of mass of all the channels' phase crossings for a given time point with weights given by the amplitude of the Hilbert envelope (Methods). Examining phase delays and the wave center path for a single trial (*Figure 2I*) revealed a pattern that resembled

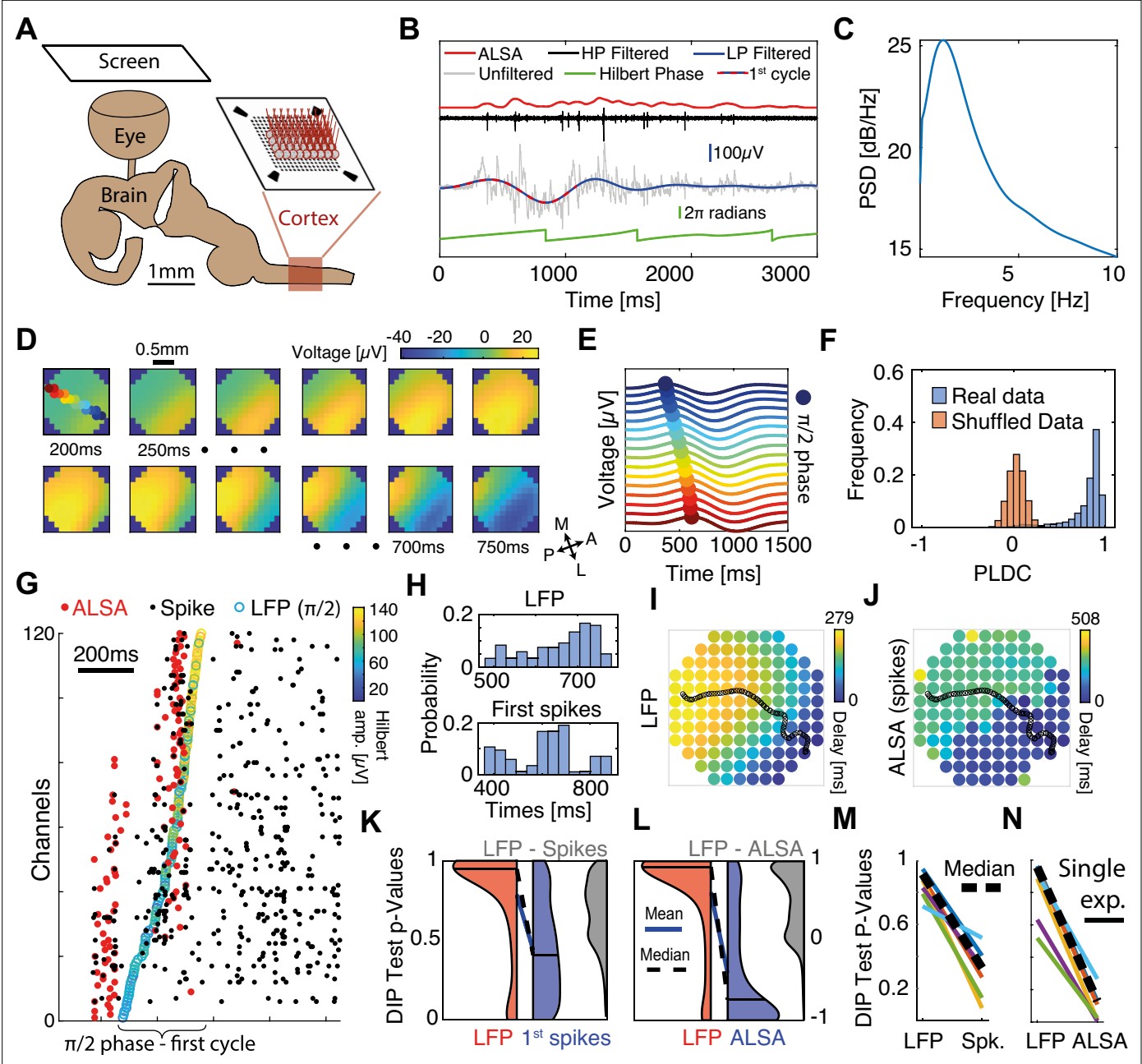

**Figure 2.** Sequential activation of discrete spiking populations underlies local field potential (LFP) waves in turtle cortex. (**A**) Schematic of the experimental setup. The brain with eye attached is extracted and the visual cortex is flattened on a multi-electrode array (MEA) and its activity is measured in response to visual stimulation of the retina. (**B**) A cortical response following visual stimulation at $t$=0. Traces show the following data: raw electrode (gray), low-pass filtered (<2 Hz, blue with dotted line marking the first oscillation cycle), high-pass filtered (>200 Hz, black), average local spiking activity (ALSA) (red), and the instantaneous phase of the low-pass filtered trace (green). (**C**) Power spectral density (averaged over 1000 randomly selected trials from one experiment). (**D**) Dynamics of low-pass filtered (<2 Hz) and averaged (n=4000) cortical responses to visual stimulation. The amplitude in each time window (200–750 ms post-stimulus in 50 ms intervals) is color-coded and presented on the physical space of the electrode. Filled color-coded circles mark the electrodes recorded in (**E**). M=medial, A=anterior, L=lateral, P=posterior. (**E**) Averaged (n=4000) low-pass filtered (<2 Hz) traces for channels along the delay gradient. The maximum is marked by a circle and its location on the array is color-coded in (**D**). (**F**) Phase latency distance correlation (PLDC) distribution for all trials with strong waves (n=2415, blue) from this experiment and for the same trials with shuffled latencies (red). (**G**) Spiking responses (black dots) in a single visual stimulation trial. Empty circles mark the $\pi/2$ phase crossing. Red circles denote the ALSA onset for each electrode. Electrodes were reordered according to the LFP phase crossings. (**H**) The distribution of LFP phase crossing times and first spike times for the response in (**G**) with corresponding DIP values (0.85 and 0.0002, respectively). (**I–J**) Latency maps on the electrode array's physical space extracted from the Hilbert phase (**I**) and ALSA (**J**). Black line denotes the wave center path extracted from the Hilbert phase (see Methods). (**K**) DIP test

*Figure 2 continued on next page*

*Figure 2 continued*

p-value distribution of the first spikes (blue), Hilbert phase crossings (red), and differences between both per trial (gray) for strong waves with enough spikes (*n*=902). Distribution medians (solid black line) are connected by a dashed black line (blue line connects the distributions' means). (**L**) Same as (**K**) but for ALSA. (**M, N**) Medians of the DIP test p-values (black lines in K, L, respectively) from six recordings in four animals. Dashed black line marks the median over all experiments. All results except (**N, M**) are from the same recording.

The online version of this article includes the following figure supplement(s) for figure 2:

**Figure supplement 1.** Identification of waves during single oscillation cycles.

**Figure supplement 2.** Phase latency maps of single trials.

**Figure supplement 3.** Spikes are locked to the 0–2 Hz oscillation.

**Figure supplement 4.** DIP statistics captures segregation of propagating activity into modules.

**Figure supplement 5.** Identification of waves for single oscillation cycles in the β-band.

the averaged propagation (*Figure 2D*) but varied between trials (*Figure 2—figure supplement 2*). We next calculated the PLDC values for the first oscillation cycle in response to visual stimulation across all trials and plotted their distribution (*Figure 2F* – blue). PLDC values were close to 1 suggesting the existence of propagating waves. For comparison, calculating PLDC after shuffling the channels' phase latencies dramatically reduced PLDC to an average close to zero (*Figure 2F* – red).

To examine whether local spike dynamics propagated continuously across space as the LFP, we examined the timing of the first spikes (marking the onset of excitability) during the first oscillation cycle of the filtered LFP. Spikes were generally locked to the LFP oscillation with a peak spiking rate at a phase of $0.53\pi$ (*Figure 2—figure supplement 3A*). To compare the spatiotemporal spiking patterns with the LFP patterns, we reordered the electrodes according to the $\pi/2$ Hilbert phase. As expected, phase gradually advanced across electrodes (*Figure 2G* – empty colored circles). In contrast, spiking activity was not continuous over space but clustered in discrete temporal bouts (*Figure 2G* – black dots). To quantify this clustering in spike times, we assigned the neurons to channels according to their maximal amplitude channel and used the DIP test to quantify temporal statistics (*Hartigan and Hartigan, 2007*). The DIP test measures the deviation of a distribution from unimodality and could well capture the differences in spike and LFP onsets. The LFP phase crossings were temporally continuous and therefore did not strongly deviate from a uniform distribution (*Figure 2H* – top, p-value 0.85). In contrast, spike timings were segregated into temporal groups and were not unimodally distributed (*Figure 2H* – bottom, p-value <0.0002). We applied this test on the first oscillation cycles across trials (*Figure 2K*). Since not all channels had active spikes in all trials, we ran the DIP test (for both spikes and LFP) only on the subset of channels with spikes within each trial. The results showed a consistently larger DIP for spikes vs. LFP (*Figure 2K* – red vs blue areas). This difference was not only significant on the average level but was also evident when comparing the p-value of pairwise differences calculated for each trial separately (*Figure 2K* – gray area). Finally, repeating this analysis for recordings in other animals reveals consistent results (*Figure 2M*), indicating that while LFP activity along the wave path resembles a continuously propagating wave, the spiking activity – a marker of local excitability – is more consistent with sequentially activated spatially segregated modules.

The above analysis was limited to the subset of channels with spikes within the first cycle in every trial. To examine propagation wavefronts across a larger fraction of channels, we constructed a more robust estimate of local spiking excitability. Specifically, we calculated the average local spiking activity (ALSA), which is the weighted average of each channel's spiking rate with its nearest neighbors' spiking rates (Methods). The spiking onset time for each channel was defined as the time of the first maximum of the ALSA (*Figure 2B* – red curve, *Figure 2G* – red dots). Since many neurons did not exhibit spiking activity during the wave onset (e.g. *Figure 2G* illustrates how for some channels, the first spikes in a cycle occur long after the activation has reached nearby channels), this measure allows estimating the wavefront propagation across many channels. Correspondingly, the ALSA phase distribution resembled that obtained with single spikes but was more sharply locked to the LFP oscillation phase (*Figure 2—figure supplement 3B*). Comparing ALSA with LFP revealed stronger differences between spikes and LFP. In contrast to the LFP (*Figure 2I*, *Figure 2—figure supplement 4A–C*), the ALSA onset times did not evolve continuously across space but were organized in two separate and sequentially activated spatial groups (*Figure 2J*, *Figure 2—figure supplement 4D–F*). Comparing the DIP test for LFP and ALSA revealed a consistent difference across trials in both the average DIP

values and the difference in single trials (*Figure 2L*) that replicated over experiments (*Figure 2N*). We note that we focused the above analysis on waves in the low-frequency LFP band (0–2 Hz) because these events recruited enough spikes to analyze their phase relations. However, propagating patterns were also detected in other frequency bands (*Figure 2—figure supplement 5*) and showed complex patterns including plane waves, radial waves, and spiral waves.

## A simple analytic model of sequentially activated Gaussians exhibits diverse WLPs

We wanted to examine whether we can capture the basic activity properties following the sequential activation of two discrete regions, using a simple analytical model. We assumed that the potential $V(x,t)$ is the sum of two one-dimensional Gaussians centered at $x_1$ and $x_2$ (with standard deviation $\sigma_x$), each multiplied by a temporal Gaussian centered around $t_1$ and $t_2$ (with standard deviation $\sigma_t$), respectively:

$$V\left(x,t\right) = \sum_{i=1,2} exp\left(-\frac{(x-x_i)^2}{2\sigma_x^2} - \frac{(t-t_i)^2}{2\sigma_t^2}\right) \tag{1}$$

In this model, the degree of signal spread (e.g. the electric field dispersion in EEG) can be modeled by changing the spatial Gaussian width, $\sigma_x$. The dynamics of these Gaussians for $x_2-x_1=\sigma_x=\sigma_t=1$ and $t_1=-1$, $t_2=1$ is plotted in *Figure 3A*. For these values, the two sequentially activated Gaussians are characterized by a WLP whose peak moves from point $x_1=-0.5$ to $x_2=0.5$. Importantly, this moving peak is always concave (*Figure 3B* – yellow curve). As the distance between Gaussians ($\Delta x=x_2 - x_1$) increases, this WLPs will look more like two discrete events and for large enough distances the peak will be replaced by a minimum (*Figure 3B* – blue curve), spatially separating the two Gaussians (*Figure 3—video 1*). To calculate the condition distinguishing between these two scenarios, we calculated the position of the propagating peak by equating the spatial derivative of *Equation 1* to zero (*Figure 3C* – black line, Appendix 1). The concavity of the propagation profile can be determined by calculating the sign of the second spatial derivative at $x=0$ and $\bar{t}=(t_1+t_2)/2$. For $\Delta 2<2\sigma_x$, this sign is negative (Appendix 1) indicating that $V(x)$ has a concave propagating peak which does not allow identifying two Gaussian peaks. This result is consistent with calculated PLDC values which are very high for $\Delta x<2\sigma_x$ (*Figure 3I*). Even for $\Delta x>2\sigma_x$, when $V(x)$ has a convex center point (*Figure 3B* – blue trace), PLDC values remain high suggesting that the values of PLDC may not be a good metric for detecting waves under these conditions. Temporal considerations can also be important for understanding the propagating maximum's dynamics. Calculating the peak dynamics at $x=0$ at times surrounding $\bar{t}$ shows that as long as the Gaussians are sequentially activated within a time frame of $\Delta t = \sqrt{2}\sigma_t$ the maximum has a single temporal peak (Appendix 1).

Given the wide range of traveling wave velocities in the literature, we calculated the velocity of WLPs' peak, $v_p$, and examined how it varies with model parameters under the condition of $\Delta x<2\sigma_x$ (Appendix 1). This derivation reveals that the velocity is maximal at $x=0$, and at that point can be described by the following equation (Appendix 1):

$$v_p = \frac{dx_p}{dt} = \frac{\Delta t}{\sigma_t^2} \cdot \frac{\sigma_x^2 \Delta x}{4\sigma_x^2 - \Delta x^2} \tag{2}$$

At small distances $\Delta x<<\sigma_x$, the velocity linearly grows with distance between Gaussians ($\Delta x$). Interestingly, as $\Delta x$ grows toward the value $2\sigma_x$, propagation velocity sharply increases (*Figure 3D*). In contrast, when the temporal overlap between the two Gaussians increases ($\Delta t$ decreases or $\sigma_t$ increases), the velocity decreases with a stronger contribution of $\sigma_t$. Thus, propagation velocity can vary depending on the distances between activated areas, their standard deviation, and the activation delay.

Traveling waves are often reported as propagating phase delays when tissue excitability is oscillatory (*Muller et al., 2018*; *Wu, 2008*). We therefore examined the spatiotemporal dynamics of two Gaussians with a temporal periodic activation profile with frequency $f$ and phase delay $\varphi$ between sources:

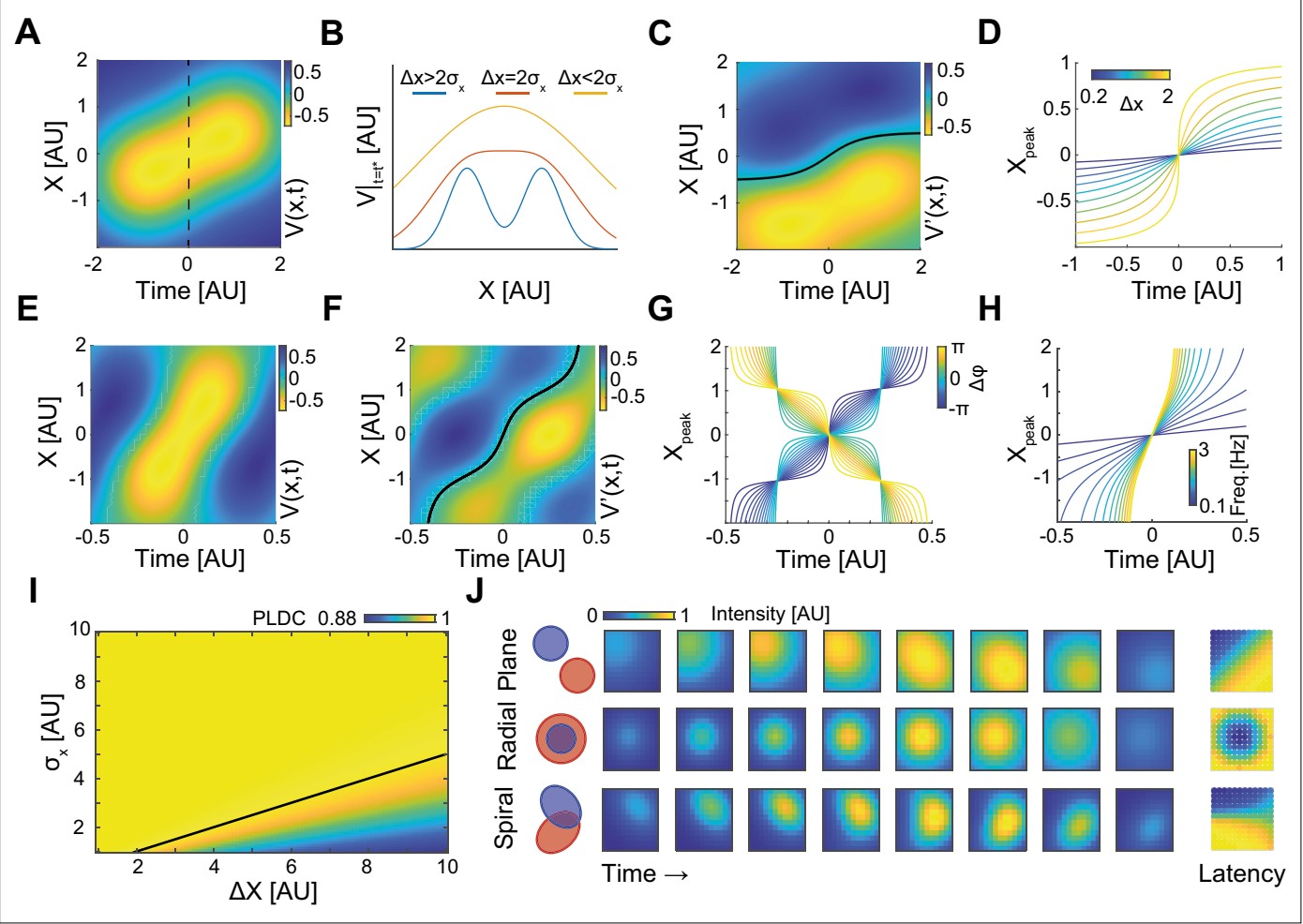

**Figure 3.** A simple analytic model of sequentially activated Gaussians exhibits traveling peaks with a diversity of patterns. (**A**) The spatiotemporal potential of two sequentially activated one-dimensional spatial Gaussians (centers; $x_1=-0.5$ and $x_2=0.5$; standard deviation $\sigma_x=1$), each with a temporal Gaussian activation profile (peaking at $t_1=-1$ and $t_2=1$; standard deviation $\sigma_t=1$), as in *Equation 1*. Notice wave-like propagation. (**B**) Spatial profile of *Equation 1* at time $t = (t_1+t_2)/2$ (dotted line in (**A**)) for three conditions. Notice that for $\Delta x \leq 2\sigma_x$ the spatial profile is concave. (**C**) The spatial derivative of $V(x,t)$ in (**A**). Black line marks Gaussian peak dynamics (zero crossing points of the spatial derivative). (**D**) Peak dynamics for different values of distances between Gaussian centers ($\Delta x$). Notice that the average velocity increases with $\Delta x$. (**E, F**) Same as in (**A, C**) but for a model of a periodically fluctuating Gaussian (*Equation 3*) ($x_1=-0.5$, $x_2=0.5$, $\sigma_x=1$, $f=1$, $\phi_1=-3\pi/8$, $\phi_2=3\pi/8$). (**G, H**) The peak dynamics (as in (**D**)) for the model in (**E, F**) as a function of the phase difference between Gaussians, $\Delta\phi$ (**G**), and $f$, the frequency of the oscillation (**H**). $f=1$ in (**G**) and $\phi_1=-\pi/4$, $\phi_2=\pi/4$ in (**H**). Notice that wave velocity increases with frequency and can change sign as a function of $\Delta\phi$. (**I**) Phase latency-distance correlation (PLDC) values calculated for different $\Delta x$ and $\sigma_x$. Black line marks the $\Delta x=2\sigma_x$ border. (**J**) Plane, radial, and spiral-like propagation patterns and the resulting phase latency maps (right) for different spatial arrangements of two sequentially activated Gaussians (left).

The online version of this article includes the following video for figure 3:

**Figure 3—video 1.** Dynamics of two consecutively activated Gaussians.

https://elifesciences.org/articles/92254/figures#fig3video1

$$V\left(x,t\right) = \sum_{i=1,2} cos\left(2\pi ft + \phi_i\right) exp\left(-\frac{\left(x - x_i\right)^2}{2\sigma_x^2}\right) \tag{3}$$

Similarly to the temporal Gaussian model, the periodic model exhibits a WLP with a traveling maximum (*Figure 3E*). Calculating the propagation velocity of the wave's peak (*Figure 3F* – black line) results in the following relation (Appendix 1):

$$v_p = 2\pi f \sin\left(\Delta\phi\right) \cdot \frac{\sigma_x^2 \left(x + \frac{\Delta x}{2}\right)^2}{\Delta x \left(x^2 + \sigma_x^2 - \left(\frac{\Delta x}{2}\right)^2\right) \exp\left(\frac{x\Delta x}{\sigma_x^2}\right)} \tag{4}$$

This equation reveals two interesting properties: First, the sign of the peak's velocity, $v_p$, depends on the phase difference between the two Gaussians ($\Delta\varphi=\varphi_2-\varphi_1$). Thus, the peak can travel either forward or backward following small changes in phase delays (*Figure 3G*). Second, the position of the peak (*Figure 3H*), and therefore the propagation speed (*Equation 4*), increases with the oscillation frequency.

The spatial activation spread ($\sigma_x$) of a cortical module can result from the measurement techniques (due to passive conductance) but also from the spatial organization of inputs and outputs patterns and their associated synaptic currents. We hypothesized that changing these patterns can produce different PLM. We therefore extended the periodic model in *Equation 3* to two dimensions (Appendix 1) and to Gaussians with different shapes (different covariance functions) (*Figure 3J*). When sequentially activated Gaussians had been placed one next to the other ($\Delta x = \sqrt{2}\sigma_x$) their propagation patterns and PLM resembled a plane wave (*Figure 3J* – top). When Gaussians overlap spatially, with the first Gaussian having a smaller $\sigma$, their sequential activation resembled a radial wave (*Figure 3J* –middle). Interestingly, calculating PLDC, which was developed to discriminate between a radial wave and a bump with increasing amplitude (*Muller and Destexhe, 2012*), resulted in a value of 0.92, indicative of a traveling wave. Finally, by positioning two Gaussians with one longer axis such that their major axes are perpendicular to one another, a pattern with features of a spiral-like wave was observed (*Figure 3J* – bottom). These examples illustrate that using a combination of only two sequentially activated Gaussians, a wide range of phase latency profiles corresponding to plane, radial, and spiral-like waves could be generated. We note that while we used simple activation profiles, spatial modules in brains (e.g. cortical columns or spatially localized projection patterns) can be more diverse and potentially facilitate a large repertoire of phase latency patterns which, despite being seemingly discrete, might be classified as traveling waves.

## Discussion

In this work we investigated the link between electrophysiological measurements of traveling waves and their underlying neuronal activations. To do so, we used common approaches for detecting traveling waves in neural data at different scales and contrasted these results with ground truth neural excitability. We showed that LFP measurements indicative of waves propagating across turtle cortex are underlined by discrete and consecutively activated neuronal populations, and not by a continuously propagating wavefront of spikes (*Figure 2*). Similarly, activation profiles that resemble continuous traveling waves in EEG simulations can be underlined by consecutive activation of two discrete cortical regions (*Figure 1*). We replicated these results using an analytical model and demonstrated that a simple scenario of sequentially activated Gaussians can exhibit WLPs with a rich diversity of spatiotemporal profiles (*Figure 3*). Our results offer insight into the scenarios and conditions for WLP detection by detecting failure points that should be considered when identifying traveling waves and therefore suggest caution when interpreting continuous PLM as microscopically propagating wave patterns. Such failure points may exist both when examining activity at the scale of brain regions (*Figure 1*) and smaller neural circuits (*Figure 2*). Therefore, our results suggest that the discrepancy between modular and wave activation should be examined across spatial scales. Specifically, it is not necessarily the case that at the fine-grained (single neuron) scale activation patterns are modular, but, following coarse graining, smooth wave patterns emerge. Rather, modular activation may hierarchically exist across scales (*Kaiser and Hilgetag, 2010*; *Meunier et al., 2010*) and may be masked by smeared spatial supra-threshold excitability boundaries. Below, we discuss these limitations across techniques and their implications.

Our analytical results suggest that the main parameter affecting the detection of WLPs in a sequentially activated modular system is the distance between modules relative to their spatial spread. In contrast to the distance between modules, the spatial spread can be strongly affected by the experimental approach used for measurements. For potential-based measurements, the spread will

markedly increase with medium conductance and distance from the source, as predicted from the volume conductance theory (*Buzsáki et al., 2012*). Our simulations show that spatial spread will be large in EEG measurements (*Figure 1*) where the distance between the sources and the electrodes is large, and the skull and meninges disperse the electrical field generated by localized cortical currents (*Akalin Acar et al., 2016*; *Kajikawa and Schroeder, 2011*). Dispersion can also be substantial in intracranial measurements such as ECoG due to the high conductivity of the cerebrospinal fluid (*Buzsáki et al., 2012*; *Herreras, 2016*). Interestingly, large spread was also observed in our MEA LFP recordings, where no meninges separate between the electrodes and tissue. Further, the small distance between the tissue and the electrodes limits conductance through the artificial cerebrospinal fluid. Despite this, propagation appeared spatially continuous in LFP, suggesting contributions from volume conduction through the tissue and non-local current sources, for example, activated synaptic terminals from axons of distant neurons. This spread is aligned with theoretical studies calculating spatial decay constants of hundreds of micrometers (*Lindén et al., 2010*) and with experiments reporting contributions from sources millimeters away (*Kajikawa and Schroeder, 2011*).

Traveling waves were also measured using VSD imaging (*Han et al., 2008*; *Muller et al., 2014*; *Senseman, 1999*; *Slovin et al., 2002*), a method that does not suffer from passive electric field spread as it measures transmembrane potentials. However, light scatters in optical signals, especially in widefield imaging. More importantly, VSD measurements report subthreshold potentials integrating excitability from large areas due to synaptic currents originating from long-distance neurons in addition to local ones (*Chemla and Chavane, 2010a*; *Petersen et al., 2003*). Correspondingly, deflection of a single whisker results in cortical activation that extends beyond the corresponding barrel (*Ferezou et al., 2006*; *Petersen et al., 2003*). From measuring VSD alone, it is unclear which component of this extended signal reflects supra-threshold activation. If supra-threshold activity is mostly restricted to the barrel and the extended spread mostly reflects sub-threshold activity, sequentially activating two nearby whiskers will result in WLPs. Similarly, sequential activation of two areas in visual space can appear to elicit a wave in the visual cortex, as observed for anesthetized cats (*Jancke et al., 2004*).

Increased spatial spread and WLP may not only result from non-local measurement but also from data analysis. For EEG specifically, amplitudes are relatively small, and consequently, waves are often reported using averaged data. In combination with the inherent variability in neuronal signals (*Fox et al., 2006*), averaging may smooth the boundaries between modules (*Mouraux and Iannetti, 2008*), effectively increase the spatial spread, and therefore promote the identification of WLPs. Such a spread may explain the smoother LFP profiles after averaging (*Figure 2D*) relative to single trials (*Figure 2G, I*) in turtle cortex.

In addition to spatial factors, temporal factors and filtering may promote WLPs. Our analytical results suggest that a single maximum will emerge when sequentially activated Gaussians considerably overlap in time ($\Delta t < \sqrt{2}\,\sigma_t$). Thus, discriminability between Gaussian activations will decrease with increase in $\sigma_t$, for example, following filtering. Indeed, filtering into narrow frequency bands is a common practice in EEG research and is liable to smoothen sharp transients. Correspondingly, our EEG simulations indicate that filtering markedly increased PLDC by effectively smoothing the activation boundaries between modules (*Figure 1—figure supplement 1C and D*). We note that the detection of WLPs is not restricted to specific frequency bands; our simulation and analytic results are not band dependent. Our turtle data also contained waves in other bands (*Figure 2—figure supplement 5*). The low spike rates in our preparation prevented us from performing a significant statistical analysis to compare these waves with underlying spiking patterns. Such examinations will be critical for understanding the propagation patterns of spikes during waves across multiple frequency bands. To do so, spatial characterization of single trials dynamics, rather than calculating average spiking locking across space (*Ray and Maunsell, 2011*), will be required.

Analysis techniques can potentially resolve the discrepancy between waves and modular activation for different levels of granularity. PLMs and PLDC analysis are valuable steps toward differentiating between single stationary bumps and traveling waves (*Muller et al., 2014*) and continue to be used to date (*Davis et al., 2020*; *Muller et al., 2018*; *Muller et al., 2016*). The underlying assumption behind this approach is that a stationary bump is characterized by a flat latency map because every channel reaches its maximum at the same time, rather than the latency increasing in a particular direction, in case of a wave. However, this approach does not consider the possibility of sequentially activated local populations. Indeed, our results show how activity profiles resulting from two sequentially

activated Gaussians meet the phase latency conditions for waves. Developing good methods for separating traveling waves from WLPs will require validating these methods using experimental paradigms providing information about spiking activity across space. In addition, more frequent usage of available techniques to decrease the spread of signals and increase localization (*Babiloni et al., 1996*) will increase the separability between waves and modular activation.

Our results can explain some of the reported variability in traveling propagation directions and velocities (*Muller et al., 2018*). We showed that the propagation direction of WLPs is determined by the location of sequentially activated modules (*Figure 1*) and the propagation speed of WLPs by the distance between modules and their temporal delay (*Figure 3*, *Equation 2*). In addition, when calculating propagation speeds for narrow-band signals, velocities increase with frequency band and may even reverse direction as a function of the phase lag between oscillatory areas (*Equation 4* and *Figure 3G–H*). These results are interesting given that cortical waves in the same tissue can propagate both in forward and backward directions and that higher frequencies propagate faster (*Bahramisharif et al., 2013*; *Halgren et al., 2019*; *Zhang et al., 2018*). Variability in propagation speed could also be detected when examining single oscillations in our turtle recordings (*Figure 2—figure supplement 1* vs *Figure 2—figure supplement 5*). The above results echo findings in a recent study showing that cortical responses to similar stimuli vary in propagation profile for EEG and MEG recordings and that this discrepancy can be accounted by modeling sequentially activated sources (*Lozano-Soldevilla and VanRullen, 2019*; *Zhigalov and Jensen, 2023*).

Variability in responses may also be explained by differences in propagation mechanisms (*Ermentrout and Kleinfeld, 2001*; *Muller et al., 2018*; *Wu, 2008*). Several reports suggest that waves are underlined by propagation along axonal collaterals (*Muller et al., 2018*; *Muller et al., 2014*). Both the transmembrane voltage-gated currents excited during action potentials as well as the post-synaptic currents along axonal boutons can potentially contribute to measured signals. However, such waves travel at high propagation speeds and are not compatible with the wide diversity of wave velocities and mechanisms of local neuronal interactions (*Ermentrout and Kleinfeld, 2001*; *Feller et al., 1996*). An intriguing possibility is that such axonal waves prime neuronal excitability by sub-threshold inputs that later result in modular supra-threshold activation. The ability to experimentally discriminate between axonal inputs and local spiking excitability (e.g. by reporters with different wavelengths) can potentially resolve such discrepancies.

Our turtle cortex results (*Figure 2*) exemplify how contrasting sub-threshold LFP measurements with supra-threshold spiking measurements can yield different conclusions about the nature of activity spread. Specifically, measuring spiking activity revealed the segregation of turtle cortex into modules. While modularity in turtle cortex connectivity structure was not systematically mapped, the temporal grouping of spike timings across two regions of turtle cortex (*Figure 2J*) is spatially aligned with a structural transition between areas D1 and D2 (*Desan, 1984*; *Prechtl et al., 1997*). This transition was not evident in the LFPs, possibly due to the longer decay constants of slow potentials and the existing connection between D2 and D1. Such connections elicit sub-threshold synaptic currents following D2 activations which can further contribute to LFP spread. The transition was also not clear when examining spiking responses in some of the trials (as indicated by high DIP scores, *Figure 2K*). However, the observation that temporal grouping became more pronounced when using ALSA (a more robust estimate of local excitability) (*Figure 2L and N*) suggests that high DIP values may result from variability in the spike times of single neurons, and not necessarily from the lack of modular activation. Such issues can be resolved by denser sampling of spiking activity in the tissue.

Similarly, sequentially activating adjacent cortical regions, which are functionally correlated and structurally connected (e.g. as activity travels across visual regions from V1 to V4), may be activated in a modular fashion but identified as traveling in VSD signals (*Gu et al., 2021*) due to mesoscale connectivity (*Oh et al., 2014*; *Yan et al., 2022*). Such modularity would be harder to detect for fast waves (in contrast to the slow waves in turtle cortex). Monitoring wave dynamics at a single cell resolution across connectivity boundaries (e.g. across two adjacent cortical regions or column) will be instrumental in identifying the granularity of propagating patterns in other cortical regions.

The above discussion calls for a better classification of dynamical phenomena into supra-threshold wave or modular excitability. In some neural systems, wave activity patterns clearly dominate. A prominent example is developing systems. Measurements from retinas ex vivo were performed across large areas with single cell resolution using both calcium imaging and dense electrode arrays (*Ackman et al.,*

*2012*; *Newman and Zahs, 1997*; *Voufo et al., 2023*; *Zheng et al., 2006*). In such measurements, spatially adjacent neurons were shown to be activated sequentially in two-dimensional spreading patterns. Traveling waves were also detected in developing cortex, thalamus, and other brain areas (*Ackman et al., 2012*; *Hanganu et al., 2006*).

In contrast, in mature circuits, and specifically in cortex, connectivity and possibly excitability profiles are replaced by modules and by segregated excitability (*Chen et al., 2008*; *Kaiser and Hilgetag, 2010*; *Meunier et al., 2010*; *Mountcastle, 1997*). Functional modular organization may result from control of neuronal excitability (*Kaiser and Hilgetag, 2010*; *Shein-Idelson et al., 2016*) in combination with modular connectivity patterns (*Meunier et al., 2010*; *Mountcastle, 1997*). Accordingly, traveling waves may dominate in specific conditions of reduced consciousness such as anesthesia or sleep (*Liang et al., 2023*; *Liang et al., 2021*; *Massimini et al., 2004*; *Muller et al., 2018*; *Xu et al., 2007*). Interestingly, waves are also reported under pathological conditions such as epilepsy (*van den Berg et al., 2012*). In such states, inhibition in the network is reduced (*Sloviter, 1987*) which may lead to a replacement of modular segregated activation by large-scale propagation of excitability crossing modular boundaries (*Shein-Idelson et al., 2016*). The abundance of traveling waves suggests that they are important components of the repertoire of neural network activity patterns. However, clear evidence for waves in some systems, for example during development, does not imply their existence in other systems. Indeed, wave propagation patterns on the single cell level are only scarcely reported during wakefulness (*Davis et al., 2020*; *Lubenov and Siapas, 2009*) leaving the microscopic correlates of most traveling wave unknown.

A promising technique for mapping the dynamics of microscopic local excitability is calcium imaging. The data provided by this method supports both localized sequential activations (*Nietz et al., 2023*) and traveling motifs (*Afrashteh et al., 2017*). However, to the best of our knowledge, no analysis was performed to discriminate between waves and sequentially activated sources and sinks. In addition, the temporal resolution of calcium imaging is low, making it hard to detect temporal boundaries, especially for fast oscillating waves (e.g. in the γ-range). Further, wide point spread functions, axonal projections, or sparse activation of adjacent populations may mask circuit modularity and promote wave detection (*Ali and Kwan, 2020*). On the other hand, the advancement of fast imaging techniques and analysis (*Zhang et al., 2023*) in combination with cell-body-targeted calcium indicators (*Shemesh et al., 2020*) and high optical sectioning methods chart a promising direction for systematic investigations into the granularity of propagating patterns in the brain. Such measurements together with new analysis approaches (*Afrashteh et al., 2017*; *Chemla et al., 2017*) will be required to disambiguate traveling waves from WLPs. Understanding the degree and the scale of functional modular organization in the brain will be important in revealing the true nature of cortical dynamics and its contribution to neural computations (*Muller et al., 2018*; *Nunez and Srinivasan, 2006*; *Pang et al., 2023*).

## Methods
### Eye-brain preparation and recording
Ex vivo experiments were performed as in *Shein-Idelson et al., 2017*, at the Max Planck Institute for Brain Research in Frankfurt and Tel Aviv University. Briefly, the brain of adult *Trachemys scripta elegans* turtles was explanted as a whole, while still attached to the eye through the optic nerve. After anterior and posterior transverse incisions, the three-layered medial and dorsal cortices, which overlay the ventricular cavity directly, were flattened and placed on top of a MEA ventricular side down (with L3 facing the MEA). 120- or 252-channel systems (MEA2100, MEA256 by Multichannel Systems, MCS) were used for recording with titanium nitrate electrode grids with 100 μm spacing and 30 μm electrode diameter (120MEA100/30iR-ITO, 256MEA100/30iR-ITO, MCS). During experiments, the tissue was constantly superfused at 10 ml/min with oxygenated turtle ACSF solution (96.5 mM NaCl, 2.6 mM KCl, 4 mM CaCl$_2$, 2 mM MgCl$_2$, 31.5 mM NaHCO$_3$, 20 mM glucose, pH 7.4) at 18°C which was bubbled with carbogen gas (95% O$_2$, 5% CO$_2$). Glutamine (0.5 mM) and HEPES (10 mM, H3375 Sigma) were added for increased retinal viability. Experiments were approved by Tel Aviv University ethical committee (Approval number: 04-18-016). Experiments and analysis were performed until further analysis yielded no new knowledge.

## Visual stimulation

The eye cup was pinned to a polydimethylsiloxane layer inside a MEA well. The visual stimulation was displayed on the screen, its size was reduced using an inverted Galilean telescope, and it was focused onto the retina using a convex lens. The stimulation position and focus on the retina were adjusted using a set of four LEDs positioned at the edges of the screen and visible on the retina. These LEDs were turned off during the experiment. Stimulations were presented using a custom MATLAB code and the psychophysics toolbox (*Brainard, 1997*). Stimulations consisted of sequentially presented full field flashes, squares flashing on the screen, natural images, or natural movies. In all stimulation, onset elicited a cortical response. Exact visual stimulation times were determined by changing the intensity of a small rectangle at the corner of the screen for every frame change and measuring this change using a photodiode.

## Spectral analysis

Spectral analysis was performed using Welch's method (*pWelch*, MATLAB). For each trial, the power spectra were extracted for a time window starting 250 ms post-stimulus and lasting 2 s. The 2 s window was divided into three 1 s blocks with 500 ms overlap.

## Hilbert transform and phase crossing

The analytic signal was calculated using MATLAB's *Hilbert* method after filtering (eighth-order Butterworth after zero padding). The signal's envelope was extracted from the norm of the analytic signal. The instantaneous phase was used to determine the oscillation's phase crossing events from the time the analytic's angle crosses a specific phase. To remove noise from phase crossing statistics, we assessed the noise level of the phase crossings. We calculated the distribution of pre-stimulus (500–1000 ms) phase crossings' amplitude of 100 random trials. Phase crossings with amplitudes lower than mean+4sd of this distribution were ignored.

## Wave detection

Phase crossings during single oscillation cycles were grouped into waves using an algorithm which identified the progression of a given phase crossing across multiple channels. We start by selecting a seed, which is a specific phase crossing event in one channel. We next go through all nearest neighboring channels. If phase crossings within a 100 ms (for 0–2 Hz band or 2 ms for 12–35 Hz band) temporal window are detected, the channel is added to the wave. This process is repeated for the added channels until no more spatiotemporally connected phase crossings are found. Only phase crossing events with Hilbert amplitudes above noise level were considered. ALSA analysis was focused on large waves recruiting many channels (2/3 for 0–2 Hz band and 5/6 for 12–35 Hz band). For first spike analysis, due to data sparseness, waves recruiting >30 channels were used.

## PLM and correlations

Maps were calculated similarly to *Muller and Destexhe, 2012*. The phase latency for each channel was defined as the time to reach a specific instantaneous phase for the ex vivo experiment, or to the first maximum for the EEG simulations. When maps were calculated on single detected waves, and a given channel had more than one phase crossing in a single cluster, only the first phase crossing was used for the latency analysis. The PLDCs for the ex vivo preparation were calculated by first calculating the Euclidean distance of all electrodes from the wave starting point (the position of the first electrode having a phase crossing in the given wave) and calculating their Pearson correlation coefficient with the phase latencies. The PLDC for the EEG simulations were taken to be the correlation of the phase latencies with the electrodes' positions on the caudal-rostral axis.

## Wave center path

WCP was calculated from the time-weighted 'center of mass' position on the electrode array for every time sample starting and ending in the first and last phase crossing, respectively. In each time sample ($t$), center of mass was calculated as follows:

$$\vec{X}_{CM}(t) = \frac{\sum_i \vec{X}_i \cdot A_i \cdot e^{\frac{-(t - t_i)^2}{2 \cdot \sigma(t)^2}}}{\sum_i A_i \cdot e^{\frac{-(t - t_i)^2}{2 \cdot \sigma(t)^2}}}$$

where the summation is over all phase crossings of a wave, $\vec{X}_i$ is the position of the $i$th phase crossing, and $A_i$ is its Hilbert amplitude. The temporal weight was taken to be a Gaussian of the time difference between the time of the $i$th phase crossing $t_i$, and the time sample for which the $X_{cm}(t)$ is calculated. To compensate for edge effects, the width of the Gaussian, $\sigma$, was varied according to progression along the wave:

$$\sigma(t) = \begin{cases} 2\overline{\Delta T} + \dfrac{8\overline{\Delta T}}{n_s/2} t & 0 \leq t < n_s/2 \\ 10\overline{\Delta T} - \dfrac{8\overline{\Delta T}}{n_s/2}(t - n_s/2) & n_s/2 \leq t < n_s \end{cases}$$

where $n_s$ is the number of samples and $\overline{\Delta T}$ is the mean temporal difference between consecutive phase crossings, calculated by averaging over all time differences between consecutive crossing times of the wave. With this scaling in $\sigma$, the samples at the beginning and end are less biased toward future or past crossings, respectively.

## Spike sorting

Spikes were sorted using a custom-designed algorithm (*Shein-Idelson et al., 2017*). After sorting, all spikes were associated with a specific channel (determined by the channel in which the averaged spike amplitude was maximal). The spiking firing pattern of each channel was comprised by merging all spikes associated with the channel.

## ALSA calculation

To calculate the ALSA, each channel's firing pattern was convolved with a constant window of 100 ms, and then smoothed using a Gaussian window of the same temporal width. The ALSA was then taken to be the spatially weighted average of these smoothed firing rates. Only the nearest neighbors were considered (up to four neighbors), with the weight of each neighbor being half the weight of the center channel. Therefore, the ALSA value of channel $i$ at time $t$ is given by:

$$ALSA(i, t) = \frac{SmoothedFireRate(i, t) + \sum_{j \in n.n.} SmoothedFireRate(j, t)/2}{1 + \sum_{j \in n.n.} 1/2}$$

The ALSA onset times were taken to be the time of the first maximum in each channel's ALSA. To reduce noise even further, only ALSA onsets that occurred around the time of the detected LFP wave were considered – from half of the wave duration before the start of the wave, to half of the wave duration after it ended.

## DIP test

The DIP test (*Hartigan and Hartigan, 2007*) is a statistic used to quantify the deviation of a distribution from unimodality. We used the MATLAB implementation by F Mechler (DIP-test; *Schluppeck, 2015a* and DIP-test significance; *Schluppeck, 2015b*). The p-value was calculated by comparing the DIP statistic to a bootstrap sample ($n=500$) of the DIP statistic calculated for a uniform probability distribution function of sample size of the empirical distribution (the uniform distribution has been shown to be the 'worst-case' unimodal distribution). Since our data is one-dimensional (first spike times, ALSA onsets, or LFP phase crossings), we use the DIP test directly on these one-dimensional distributions. In DIP analysis of first spikes, the following exclusion criteria were pre-established. We only considered large enough waves (having at least 5/6 channels participating in the LFP wave). In addition, we only examined the first spikes (marking activity onset) and therefore only analyzed channels with spikes occurring during the waves (i.e. during the interval $[T-W, T+W]$ where $T$ is each wave's

middle time and *W* is its width). When running the analysis on ALSA, were analyzed all trials with at least 4/6 channels.

## EEG simulations

Simulations were performed using Brainstorm (version 3.220616) in MATLAB (*Tadel et al., 2011*). Electrical field and consequent EEG recordings flowing defined activation profiles were simulated based on a human anatomical MRI scan. Activity sources were synchronized groups of dipoles located at the cortex surface and oriented normally to it. These grouped dipoles (red and blue dots in *Figure 1B*) are called scouts in the Brainstorm jargon and were manually marked on the cortical surface using the Brainstorm GUI. Each scout is comprised of 160 adjacent dipoles and has an average area of 27 cm$^2$ (20 dipoles of area 0.187 cm$^2$ for small scouts). In each simulation, two scouts were activated with the same temporal Gaussian function (amplitude 1; standard deviation [sd] 0.2 s). This Gaussian was repeated consecutively five times with a period of 8sd, simulating activity for 580 ms (5800 samples; 10 kHz sampling rate). To simulate activation lags between modules (as in *Figure 1*), the temporal mean of one of the scouts was shifted by 0.2sd in every independent simulation, resulting with temporal differences ranging from 0.2sd to 2sd. In the same fashion, we repeated the simulation for varying spatial distance between the scout's centers (along the posterior-anterior axis) divided by the standard deviation. The distances were the Euclidean distance between the scouts' centers (mean position of each scouts' dipoles), and the standard deviation was calculated along the posterior-anterior axis for each scout (the mean of the two scouts was taken as the final value). In Brainstorm we used the default MRI anatomy (ICBM_152), with 15,002 vertices (or 306,716 for small scouts) and the 'ICBM152' -> 'ACA 10-05 343' EEG headset device (with 343 electrodes). We only analyzed the activity of the top 63 electrodes (35 for small scouts) at the vicinity of the sources. In the small scouts simulation (*Figure 1—figure supplement 1A and B*), we duplicated the sources such that they are symmetric across the two hemispheres. We modeled the electromagnetic properties of the head and the EEG device (forward model) using the OpenMEEG BEM method (*Gramfort et al., 2010*; *Kybic et al., 2005*) without inserting noise to the process (identity noise covariance matrix).

## Acknowledgements

This project has received funding from the Joint ISF-NSFC Research Fund (grant no. 3430/20).

## Additional information

### Funding

| Funder | Grant reference number | Author |
| --- | --- | --- |
| Israel Science Foundation | ISF-NSFC Research Fund 3430/20 | Mark Shein-Idelson |
| National Natural Science Foundation of China | ISF-NSFC Research Fund 3430/20 | Mark Shein-Idelson |

The funders had no role in study design, data collection and interpretation, or the decision to submit the work for publication.

### Author contributions

Yuval Orsher, Conceptualization, Software, Formal analysis, Validation, Investigation, Visualization, Methodology, Writing – original draft; Ariel Rom, Formal analysis, Validation, Investigation, Methodology; Rotem Perel, Investigation, Methodology; Yoav Lahini, Pablo Blinder, Funding acquisition, Writing – review and editing; Mark Shein-Idelson, Conceptualization, Formal analysis, Supervision, Funding acquisition, Investigation, Visualization, Methodology, Writing – original draft, Writing – review and editing

### Author ORCIDs

Yuval Orsher ⓘ https://orcid.org/0009-0007-0465-7199

Yoav Lahini [ID] http://orcid.org/0000-0002-5952-7144
Pablo Blinder [ID] http://orcid.org/0000-0002-4042-214X
Mark Shein-Idelson [ID] http://orcid.org/0000-0001-8132-2826

### Ethics

Experiments were approved by Tel Aviv University ethical committee (Approval number: 04-18-016).

Joint Public Review: https://doi.org/10.7554/eLife.92254.3.sa1
Author Response https://doi.org/10.7554/eLife.92254.3.sa2

## Additional files

### Supplementary files

• MDAR checklist

### Data availability

All the data presented in this study, including individual data points, are included in the figures, or can be downloaded from Zenodo.

The following dataset was generated:

| Author(s) | Year | Dataset title | Dataset URL | Database and Identifier |
|---|---|---|---|---|
| Orsher Y | 2023 | EvolutionaryNeuralCodingLab/ Travelling-waves-or-sequentially-activated-discrete-modules: eLife publication | https://zenodo.org/ records/10429410 | Zenodo, 10.5281/ zenodo.10429410 |

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

## Appendix 1

### Theoretical model

Sequential activation of modules was represented by the sum of two one-dimensional Gaussians centered at $x_1$ and $x_2$ and multiplied by a temporal Gaussian centered around $t_1$ and $t_2$, respectively:

$$V(x,t) = \sum_{i=1,2} \exp\left(-\frac{(x-x_i)^2}{2\sigma_x^2} - \frac{(t-t_i)^2}{2\sigma_t^2}\right)$$

with spatial and temporal widths $\sigma_x, \sigma_t$. By moving to center coordinates: $\bar{x} = \frac{x_1+x_2}{2}$, $\Delta x = x_2 - x_1$, $\bar{t} = \frac{t_1+t_2}{2}$, $\Delta t = t_2 - t_1$ and setting the average position to zero we get:

$$V(x,t) = \sum_{i=1,2} \exp\left(-\frac{\left(x-(-1)^i \Delta x/2\right)^2}{2\sigma_x^2} - \frac{\left(t-\left(\bar{t}+(-1)^i \Delta t/2\right)\right)^2}{2\sigma_t^2}\right)$$

To calculate the time at which the wave passes through the midpoint between Gaussians, we calculate the temporal derivative at $x=0$ and equate it to zero. This yields the following relation which is satisfied at $t=\bar{t}$.

$$\frac{\left(t-(\bar{t}-\Delta t/2)\right)}{\left(t-(\bar{t}+\Delta t/2)\right)} = -\exp\left(\frac{\Delta t(t-\bar{t})}{\sigma_t^2}\right)$$

To track the wavefront's peak, we calculate the spatial extremum by equating the first spatial derivative to zero at $t=\bar{t}$.

$$\frac{(x+\Delta x/2)}{(x-\Delta x/2)} = -\exp\left(\frac{x\Delta x}{\sigma_x^2}\right)$$

This equation is satisfied at $x=0$. We calculate the second derivative to determine whether the extremum is concave or convex.

$$\frac{\partial^2 V}{\partial x^2}\big|_{t=\bar{t},x=0} = \frac{2}{\sigma_x^2}\exp\left(-\frac{(\Delta x/2)^2}{2\sigma_x^2} - \frac{(\Delta t/2)^2}{2\sigma_t^2}\right)\left(\frac{1}{\sigma_x^2}(\Delta x/2)^2 - 1\right)$$

Since the sign of this expression depends only on the last multiplicand, we conclude that the extremum at $t=\bar{t}, x=0$ will be a maximum when $\Delta x < 2\sigma_x$ and a minimum for when $\Delta x > 2\sigma_x$.

Similarly to the spatial domain, we can calculate the conditions for identifying two Gaussians over the temporal domain. We require that no minimum appears at $x=0$ during the period between the two peaks reaching their maximum, i.e., $\forall t \in (t_1, t_2)$, $\frac{\partial^2 V}{\partial t^2}(x=0,t) \leq 0$. This results in the following equation:

$$\frac{\left(\left(t-\left(\bar{t}-\frac{\Delta t}{2}\right)\right)^2 - \sigma_t^2\right)}{\left(\left(t-\left(\bar{t}+\frac{\Delta t}{2}\right)\right)^2 - \sigma_t^2\right)} \leq -\exp\left(\frac{(t-\bar{t})\Delta t}{\sigma_t^2}\right)$$

By separately considering the case of $t \geq \bar{t}$ and $t < \bar{t}$, one can show that:

$$(t-\bar{t})^2 + \left(\frac{\Delta t}{2}\right)^2 \leq \sigma_t^2 \quad \forall t \in (t_1, t_2)$$

Plugging in $t=t_1$ or $t=t_2$ (in which the left-hand side is maximal) yields a tight bound of $\Delta t \leq \sqrt{2}\sigma_t$.

To calculate the velocity profile of the peak, we find $\frac{\partial V}{\partial x}$ and equate it to zero. This yields the following relation between $x_p$ and $t$, depicting the coordinates in which the profile is at its peak:

$$\left(x_p + \frac{\Delta x}{2}\right) \exp\left(-\frac{x_p \Delta x}{\sigma_x^2} + \frac{\Delta t \left(\bar{t} - t\right)}{\sigma_t^2}\right) = -\left(x_p - \frac{\Delta x}{2}\right)$$

From this equation, one can isolate $t = g(x_p)$

$$\bar{t} - \frac{\sigma_t^2}{\Delta t} \ln\left[-\frac{\left(x_p - \frac{\Delta x}{2}\right)}{\left(x_p + \frac{\Delta x}{2}\right)}\right] - x_p \frac{\Delta x}{\Delta t} \frac{\sigma_t^2}{\sigma_x^2} = t$$

From this equation we calculate $\frac{dt}{dx_p}$, and under the condition that $\Delta x < 2\sigma_x$, we can use the inverse function theorem to get $\frac{dx_p}{dt}$:

$$\frac{dx_p}{dt} = \frac{\Delta t}{\sigma_t^2} \cdot \frac{\sigma_x^2 \Delta x}{4\sigma_x^2 - \Delta x^2}$$

A similar approach was taken for calculating the velocity of the peak for an oscillating sources model:

$$V\left(x, t\right) = \sum_{i=1,2} \cos\left(2\pi f t - \phi_i\right) \cdot \exp\left(-\frac{\left(x - x_i\right)^2}{2\sigma_x^2}\right)$$

To calculate the peak velocity, we change coordinates to $\bar{x} = \frac{x_1 + x_2}{2}$, $\Delta x = x_2 - x_1$, $\Delta \phi = \phi_1 - \phi_2$, and equate the spatial derivative to zero:

$$\frac{\partial V}{\partial x} = -\frac{1}{\sigma_x^2} \sum_{i=1,2} \cos\left(2\pi f t - \phi_i\right) \cdot \left(x - (-1)^i \frac{\Delta x}{2}\right) \exp\left(-\frac{\left(x - (-1)^i \frac{\Delta x}{2}\right)^2}{2\sigma_x^2}\right) = 0$$

Giving rise to the condition:

$$\frac{\cos\left(2\pi f t - \phi_1\right)}{\cos\left(2\pi f t - \phi_2\right)} = -\frac{\left(x - \frac{\Delta x}{2}\right)}{\left(x + \frac{\Delta x}{2}\right)} \exp\left(\frac{x \Delta x}{\sigma_x^2}\right)$$

We next calculate the partial derivative with respect to $t$, to get:

$$\frac{dx}{dt} = 2\pi f \sin\left(\Delta \phi\right) \cdot \frac{\sigma_x^2 \left(x + \frac{\Delta x}{2}\right)^2}{\Delta x \left(x^2 + \sigma_x^2 - \left(\frac{\Delta x}{2}\right)^2\right) \exp\left(\frac{x \Delta x}{\sigma_x^2}\right)}$$

For examining PLM under different spatially distributed Gaussians, we used a simplistic representation in which two two-dimensional spatial Gaussians were each multiplied by a temporal squared sine:

$$V\left(x, y, t\right) = \sum_{i=1,2} w\left(t - t_i\right) \sin^2\left(t - t_i\right) \exp\left(-\frac{\left(x - x_i\right)^2 + \left(y - y_i\right)^2}{2\sigma_i^2}\right)$$

where $w = \Theta(t)\Theta(\pi - t)$ is a window function ($\Theta$ is the Heaviside step function) and $t_i$ is the time in which pulse $i$ starts. For generating phase WLPs with different spatial arrangements, we used

the following parameters. Plane wave: $t_1 = 1; t_2 = 30$ (70% temporal overlap); $(x_1, y_1) = (3, 3)$ ; $(x_2, y_2) = (9, 9)$ ; and $\sigma_1 = \sigma_2 = 6$ ; $\Delta x = \sqrt{2}\sigma$. Radial wave: $t_1 = 1; t_2 = 30$; $(x_1, y_1) = (x_2, y_2) = (3, 3)$; $\sigma_1 = 3; \sigma_2 = 6$. For the spiral-like wave we used two Gaussians with non-zero covariance:

$$V\left(\vec{X}, t\right) = \sum_{i=1,2} w\left(t - t_i\right) \sin^2\left(t - t_i\right) \exp\left(-\frac{1}{2}\left(\vec{X} - \vec{X_i}\right)^T \Sigma_i^{-1} \left(\vec{X} - \vec{X_i}\right)\right)$$

with $\vec{X}$ being the spatial coordinates, $\vec{X_i}$ the center of the $i$th Gaussian, and $\Sigma_i$ is its covariance matrix. Parameters were $t_1 = 1; t_2 = 30$; $\vec{X}_1 = (8, 4)$; $\vec{X}_2 = (8, 8)$; $\Sigma_i = R^{-1}\left(\alpha_i\right) GR\left(\alpha_i\right)$ where $R = \begin{pmatrix} \cos\left(\alpha\right) & -\sin\left(\alpha\right) \\ \sin\left(\alpha\right) & \cos\left(\alpha\right) \end{pmatrix}$ is the rotation matrix, $G = \begin{pmatrix} 8 & 0 \\ 0 & 15 \end{pmatrix}$ is the covariance matrix of a vertical elliptical Gaussian, and $\alpha_{1,2} = \pm\frac{\pi}{4}$ .

