## [Editor Report · eLife assessment]

In this **important** work, authors show that brain activity thought to be a travelling wave may just be a series of sequentially activated sources at the neuron spiking level. They support this with **convincing** results from a turtle cortex preparation and relevant simulations. This work will be of interest to neuroscientists interested in understanding how cortical computations are made.

---

## [Referee Report · Joint Public Review]

Summary:

In this interesting work, the authors investigated an important topical question: when we see travelling waves in cortical activity, is this due to true wave-like spread, or due to sequentially activated sources? In simulations, it is shown that sequential brain module activation can show up as a travelling wave - even in improved methods such as phase delay maps - and a variety of parameters is investigated. Then, in ex-vivo turtle eye-brain preparations, the authors show that visual cortex waves observable in local field potentials are in fact often better explained as areas D1 and D2 being sequentially activated. This has implications for how we think about travelling wave methodology and relevant analytical tools.

Strengths:

I enjoyed reading the discussion. The authors are careful in their claims, and point out that some phenomena may still indeed be genuine travelling waves, but we should have a higher evidence bar to claim this for a particular process in light of this paper and Zhigalov & Jensen (2023) (ref 44). Given this careful discussion, the claims made are well-supported by the experimental results. The discussion also gives a nice overview of potential options in light of this and future directions.

The illustration of different gaussian covariances leading to very different latency maps was interesting to see.

Furthermore, the methods are detailed and clearly structured and the Supplementary Figures, particularly single trial results, are useful and convincing.

---

## [Author Response]

The following is the authors’ response to the original reviews.

Summary:In this interesting work, the authors investigated an important topical question: when we see travelling waves in cortical activity, is this due to true wave-like spread, or due to sequentially activated sources? In simulations, it is shown that sequential brain module activation can show up as a travelling wave - even in improved methods such as phase delay maps - and a variety of parameters is investigated. Then, in ex-vivo turtle eye-brain preparations, the authors show that visual cortex waves observable in local field potentials are in fact often better explained as areas D1 and D2 being sequentially activated. This has implications for how we think about travelling wave methodology and relevant analytical tools.Strengths:I enjoyed reading the discussion. The authors are careful in their claims, and point out that some phenomena may still indeed be genuine travelling waves, but we should have a higher evidence bar to claim this for a particular process in light of this paper and Zhigalov & Jensen (2023) (ref 44). Given this careful discussion, the claims made are well-supported by the experimental results. The discussion also gives a nice overview of potential options in light of this and future directions.The illustration of different gaussian covariances leading to very different latency maps was interesting to see.Furthermore, the methods are detailed and clearly structured and the Supplementary Figures, particularly single trial results, are useful and convincing.

We are glad the reviewer found our manuscript “interesting”, the questions we raise “important”, our claims “well-supported by the experimental results”, and our methods “detailed and clearly structured”.

The details of the sequentially activated Gaussian simulations give some useful results, but the fundamental idea still appears to be "sequential activation is often indistinguishable from a travelling wave", an idea advanced e.g. by Zhigalov & Jensen (2023). It takes a while until the (in my opinion) more intriguing experimental results.

To emphasize the experimental results, we switched between the analytical results and the experimental results. Correspondingly, figure 2 now illustrates the more intriguing experimental results and figure 3 the analytical results. In addition, we added subtitles to the different sections of the results to ease the navigation through the paper and to enable the readers to access the different sections more easily.

One of the key claims is that the spikes are more consistent with two sequentially activated modules rather than a continuous wave (with Fig 3k and 3l key to support this). Whilst this is *more* consistent, it is worth mentioning that there seems to be stochasticity to this and between-trial variability, especially for spikes.

In the revised manuscript we added the reviewer’s comment about stochasticity, and we discuss its possible origins:

"The transition was also not clear when examining spiking responses in some of the trials (as indicated by high DIP scores, Figure 2K). However, the observation that temporal grouping became more pronounced when using ALSA (a more robust estimate of local excitability) (Figure 2L,N), suggests that high DIP values may result from variability in the spike times of single neurons, and not necessarily from the lack of modular activation. Such issues can be resolved by denser sampling of spiking activity in the tissue."

**Recommendations For The Authors:**
The eye-cortex turtle preparation is not the most common. I would add more context about how specific the results are to this preparation vs how comparable it is to human data.

We added a sentence explaining the relevance of our preparation: “Finally, while the layered organization of turtle cortex is different than that of mammalian cortex, the basic excitability features of both tissues are similar (Connors and Kriegstein, 1986; Hemberger et al., 2019; Kriegstein and Connors, 1986; Larkum et al., 2008; Shein-Idelson et al., 2017b), and substantial differences in the manner by which field potentials and spikes spread through the tissue are not to be expected.”

Philosophical question: when does a 'module' become small enough for it to count as a travelling wave? More on this could be added to the discussion. I think we are in the very early days for a true understanding of travelling waves, and I wonder if these sequentially activated modules will functionally correspond to the known cortical segregation, or if it varies by area/task.

We agree with the reviewer that macroscopic waves could be composed of smaller modules (or single neurons at the smallest scale). Our results suggest that modular patterns can be classified as wave patterns both at large scales (of brain areas) and smaller scales of local neural circuits. Therefore, we believe it is necessary to make this distinction across different scales. We sharpened this point in the first paragraph of the discussion:

"…We showed that LFP measurements indicative of waves propagating across turtle cortex are underlined by discrete and consecutively activated neuronal populations, and not by a continuously propagating wavefront of spikes (Figure 2). Similarly, activation profiles that resemble continuous travelling waves in EEG simulations can be underlined by consecutive activation of two discrete cortical regions (Figure 1). We replicated these results using an analytical model and demonstrated that a simple scenario of sequentially activated Gaussians can exhibit WLPs with a rich diversity of spatiotemporal profiles (Figure 3). Our results offer insight into the scenarios and conditions for WLP detection by identifying failure points that should be considered when identifying travelling waves and therefore suggest caution when interpreting continuous phase latency maps as microscopically propagating wave patterns. Such failure points may exist both when examining activity at the scale of brain regions (Figure 1) and smaller neural circuits (Figure 2). Therefore, our results suggest that the discrepancy between modular and wave activation should be examined across spatial scales. Specifically, it is not necessarily the case that at the fine grained (single neuron) scale activation patterns are modular, but, following coarse graining, smooth wave patterns emerge. Rather, modular activation may hierarchically exist across scales (Kaiser and Hilgetag, 2010; Meunier et al., 2010) and may be masked by smeared spatial supra-threshold excitability boundaries. Below we discuss these limitations across techniques and their implications.”

I would advise the authors to focus on the experimental data, perhaps by putting the simulations second, and by putting some of the equation details that are in Methods into the Supplementary Information. Whilst the simulation parameter space is well-explored, the fundamental idea of spreading Gaussians is relatively simple, and the current manuscript organization detracted from the main message for me a little bit.”

Following the referee’s suggestion, we switched between the section with experimental data and the one with the analytic model (see response to comment 1). In addition, to ease the reading of the methods, we moved the mathematical derivation and related equations to appendix 1.

Things I thought about that you may also enjoy thinking about:Could we tell something about sequential sources vs travelling waves by the nature of the wave - e.g. shape or dispersion? If some wave properties are conserved whilst travelling, this could be evidence for travelling vs two sources.

This is a wonderful suggestion. We are currently working on a follow up publication with a new approach to do exactly that! We think that this new body of work is outside the scope of this paper.

Could synaptic potentials spread like waves, but spikes more in modular bursts? This would also explain the LFP vs spikes difference - maybe travelling waves of EPSPs are there priming the network, 'looking' for suitable modules to activate, which then activate sequentially. The current discussion is quite spike-focused - could some information be in synaptic potentials after all?

This is an interesting idea with intriguing functional implications. We added this idea to our discussion (see paragraph below). In addition, to emphasize our discussion on synaptic potentials, we reorganized the paragraphs in the discussion to separate between our discussion on sub-threshold excitability (which is mostly synaptic) and supra-threshold excitability which is the focus of the second part of the discussion.

“Variability in responses may also be explained by differences in propagation mechanisms (Ermentrout and Kleinfeld, 2001; Muller et al., 2018; Wu et al., 2008). Several reports suggest that waves are underlined by propagation along axonal collaterals (Muller et al., 2018, 2014). Both the transmembrane voltage-gated currents excited during action potentials as well as the post-synaptic currents along axonal boutons can potentially contribute to measured signals. However, such waves travel at high propagation speeds and are not compatible with the wide diversity of wave velocities and mechanisms of local neuronal interactions (Ermentrout and Kleinfeld, 2001; Feller et al., 1996). An intriguing possibility is that such axonal waves prime neuronal excitability by sub-threshold inputs that later result in modular supra-threshold activation. The ability to experimentally discriminate between axonal inputs and local spiking excitability (e.g. by reporters with different wavelengths) can potentially resolve such discrepancies.

Our turtle cortex results (Figure 2) exemplify how contrasting sub-threshold LFP measurements with supra-threshold spiking measurements can yield different conclusions about the nature of activity spread….”